# Genetic Diversity and Population Structure in *Solanum nigrum* Based on Single-Nucleotide Polymorphism (SNP) Markers

Jinhui Li [1,†], Shouhui Wei [1,†] ⓘ, Zhaofeng Huang [1], Yuyong Zhu [2], Longlong Li [1], Yixiao Zhang [1], Ziqing Ma [1] and Hongjuan Huang [1,*]

1   State Key Laboratory for Biology of Plant Diseases and Insect Pests, Institute of Plant Protection, Chinese Academy of Agricultural Sciences, Beijing 100193, China
2   General Station of Agriculture Technology Extension, Xinjiang Production and Construction Corps, Urumqi 830011, China
*   Correspondence: hjhuang@ippcaas.cn; Tel.: +86-(10)-62815937
†   These authors contributed equally to this work.

**Abstract:** *Solanum nigrum* is a noxious weed in agricultural ecosystem that limits many crops' production. The aim of the present study was to understand the level of genetic diversity and population structure of *S. nigrum* in China. A specific-locus amplified fragment (SLAF) sequencing method was conducted to detect single-nucleotide polymorphisms (SNPs) in the genomes of *S. nigrum* from 66 populations in China. A total of 616,533 high-quality SNPs were identified from 189,840 SLAFs, with an average sequencing depth of $10.59\times$ fold and a Q30 value of 93% and a GC content of 42.78%. It showed a considerable amount of genetic diversity and genetic variability of *S. nigrum* among samples. The genetic differentiation of *S. nigrum* indicated that there was a low level of genetic differentiation (Fst < 0.1000) among geographical populations. A cluster analysis showed that populations of *S. nigrum* were divided into two subgroups, with some samples from adjacent position roughly clustered together, which showed some correlation between geographic origins. A population structure analysis suggested the 66 *S. nigrum* samples could have originated from three different genetic clusters. The Xinjiang site was the only location where all genetic clusters were found, which suggested these populations were genetically diverse. These results showed that there was a high degree of genetic diversity and low difference among the different geographical populations of *S. nigrum*. The results from the genetic structure of the SNP markers indicated that wide genetic variability exists among the population of *S. nigrum* in China, which may contribute to the adaptation and infestation of this weed species.

**Keywords:** *Solanum nigrum* L.; single-nucleotide polymorphisms (SNPs); specific-locus amplified fragment (SLAF); genetic diversity; population structure

## 1. Introduction

Black nightshade (*Solanum nigrum* L.) is widely distributed throughout the world. It serves as a source of leafy vegetables, fruits, or local medicinal herbs [1–3]. However, this species is considered to be a troublesome weed of agriculture in most parts of the world. *S. nigrum* causes substantial problems for crop production not only by competing for nutrients, moisture, and light with crops in an agricultural field, but also by reducing the commercial quality of crops by contaminating harvest crops because of their purple/black succulent berries. It also widespread in China, especially in the northernmost parts of China, abundant in irrigated fields in Xinjiang, Heilongjiang, and Jilin. In recent years, persistent herbicide application has resulted in herbicide-resistant *S. nigrum* populations in China [4,5]. It has become a noxious weed of crops such as cotton, corn, potato, soybean, sunflower, tomato, and vegetables, especially in cotton fields since the land is subject to irrigation in Xinjiang [4–7]. Based on our survey, more than half of cotton fields were infested with *S. nigrum*, and the number of *S. nigrum* has increased dramatically compared

with a decade ago [6]. The rapid increase in the number and spread of *S. nigrum* makes it the dominant weed in the cotton production area.

Populations of *S. nigrum* are rapidly expanding due to adapting to the environment. Genetic diversity influences populations' adaptation to cropping systems and species evolution. *S. nigrum* has accumulated many characteristics and genetic adaptations to the environment. It possesses prolific seed production, wide tolerant habits from tropical to temperate regions [1], the ability to accumulate Cd from soil [8,9], and resistance to serval herbicides [4,5,10–13]. Several PCR-based DNA markers, namely restriction fragment length polymorphism (RFLP), nuclear intron-targeting markers, and inter-sample sequence repeats (SSR), have been used to discover genetic diversity and variation among *S. nigrum* populations [14–17]. However, information on the genetic polymorphism among different *S. nigrum* populations in China has not been studied. Assessment of its genetic diversity will be helpful for better understanding the genetic structure of natural *S. nigrum* populations. An effective weed management program requires a comprehensive knowledge of the weed diversity [18–20].

Molecular markers, reflecting the actual level of genetic variation at the DNA level, have been used to estimate genetic diversity in plants because they are stable, polymorphic, readily available in the genome, and not sensitive to environmental factors [21]. SNP markers have been rapidly applied in genetics because of the abundance and accessibility of nonbiased SNPs throughout the genome [22]. They have been widely used in plants to evaluate genetic diversity, construct linkage maps, and perform association analyses [23,24]. The SLAF-seq method combining next-generation sequencing was developed for genetic mapping, polymorphism analysis, systematic evolution, and germ plasm resource identification [25–29].

In the present paper, SLAF-seq technology was used for the first time to seek molecular evidence of *S. nigrum*. Here we report on the development of SNP markers for examining the underlying population structure of *S. nigrum* based on 66 samples from a wide range in China. Based on the wide distribution and rapid increase in the number of *S. nigrum*, genetic diversity and possible differentiation within *S. nigrum* populations are hypothesized. In this study, we aimed to assess the genetic diversity and genetic structure in natural populations of *S. nigrum* by comparing the genetic distance and the number of polymorphism markers. This is the first report to use SNPs on *S. nigrum* and represent whether genetic diversity is randomly distributed or has a patterned spatial distribution.

## 2. Materials and Methods

### 2.1. Plant Materials

A total of 66 seed samples of *S. nigrum* was collected from crop fields. These seeds were obtained from different locations ranging from 34° N to 52° N and from 80° E to 133° E in China, and at elevations ranging from 50 m to 780 m (Table 1). Of these samples, 47 were from northwest China (Xinjiang), 15 were from northeast China (Heilongjiang, Liaoning, Jilin, and Inner Mongolia), 4 were from the north of China (Shaanxi, Henan, and Beijing). Ripe seeds were collected from five to ten individual plants randomly. The *S. nigrum* were cultivated in the greenhouse of the Institute of Plant Protection, Chinese Academy of Agricultural Sciences, China.

### 2.2. Genomic DNA Extractions and Specific-Locus Amplified Fragment Sequencing

Genomic DNA was isolated from 100 mg of fresh young leaf tissue from germinated seeds at the 3–4-leaf stage following the CTAB manufacturer's protocol. The DNA quality was then assessed using electrophoresis on a 1.2% agarose gel. A high-resolution technology named SLAF-seq was constructed for DNA sequencing as previously described [30], except that the genomic DNA was digested with the restriction enzyme HaeIII (New England Biolabs, NEB). In summary, the genomic DNA was digested into fragments with the restriction enzyme HaeIII with a size-selection window of 364–414 bp and then sequenced with the HiSeq 2500 system according to the manufacturer's protocol (Illumina Inc., San

Diego, CA, USA). It was expected to yield 113,951 unique SLAF tags. To control the quality, we screened the raw data for high-quality reads with Q30 (the ratio of bases sequencing quality value greater than 30) quality scores and guanine/cytosine content for reads 94.11% and 42.78%, respectively (Table 2). The genome size of *S. nigrum* was estimated to be 1.66 G using flow cytometry.

**Table 1.** Details of geographic and sampling information for *S. nigrum* populations in this study.

| ID | Biogeographic Regions | Longitude | Latitude | ID | Biogeographic Regions | Longitude | Latitude |
|---|---|---|---|---|---|---|---|
| SB1 | Beijing | 116°16′45″ | 39°33′16″ | SX15 | Xinjiang | 85°63′07″ | 44°19′31″ |
| SH1 | Heilongjiang | 122°77′19″ | 52°22′92″ | SX16 | Xinjiang | 86°50′25″ | 44°23′50″ |
| SH2 | Heilongjiang | 133°52′53″ | 47°24′19″ | SX17 | Xinjiang | 84°58′38″ | 44°34′17″ |
| SH3 | Heilongjiang | 124°54′42″ | 47°08′55″ | SX18 | Xinjiang | 86°50′25″ | 44°23′50″ |
| SH4 | Heilongjiang | 124°28′01″ | 47°45′39″ | SX19 | Xinjiang | 85°0′12″ | 44°27′45″ |
| SJ1 | Jilin | 123°43′53″ | 44°10′83″ | SX20 | Xinjiang | 86°37′30″ | 44°28′59″ |
| SJ2 | Jilin | 124°1′57″ | 44°17′20″ | SX21 | Xinjiang | 81°40′59″ | 45°0′14″ |
| SJ3 | Jilin | 122° 55′82″ | 45°07′49″ | SX22 | Xinjiang | 86°24′17″ | 44°30′58″ |
| SJ4 | Jilin | 122°54′14″ | 44°36′12″ | SX23 | Xinjiang | 85°1′36″ | 44°31′37 |
| SJ5 | Jilin | 123°55′52″ | 44°17′31″ | SX24 | Xinjiang | 80°54′50″ | 40°34′57″ |
| SJ6 | Jilin | 123°46′35″ | 44°13′60″ | SX25 | Xinjiang | 81°18′6″ | 40°32′37″ |
| SJ7 | Jilin | 123°48′56″ | 44°41′54″ | SX26 | Xinjiang | 89°08′31″ | 42°57′14″ |
| SL1 | Liaoning | 120°50′50″ | 40°44′18″ | SX27 | Xinjiang | 86°33′85″ | 46°19′43″ |
| SL2 | Liaoning | 120°52′35″ | 40°46′51″ | SX28 | Xinjiang | 85°42′23″ | 46°05′11″ |
| SM1 | Inner Mongolia | 121°19′52″ | 43°45′52″ | SX29 | Xinjiang | 84°43′12″ | 45°25′26″ |
| SM2 | Inner Mongolia | 121°19′52″ | 43°45′52″ | SX31 | Xinjiang | 84°37′17″ | 44°32′21″ |
| SN1 | Henan | 113°41′44″ | 35°15′32″ | SX32 | Xinjiang | 84°58′38″ | 44°34′17 |
| SN2 | Henan | 114°55′22″ | 34°51′59″ | SX33 | Xinjiang | 82°06′20″ | 44°58′02″ |
| SS1 | Shaanxi | 108°13′57″ | 34°23′38″ | SX34 | Xinjiang | 85°36′49″ | 48°08′38″ |
| SX02 | Xinjiang | 81°32′6″ | 45°5′12″ | SX35 | Xinjiang | 84°59′56″ | 44°25′45″ |
| SX03 | Xinjiang | 86°24′17″ | 44°30′58″ | SX36 | Xinjiang | 86°37′30″ | 44°28′59″ |
| SX04 | Xinjiang | 82°20′33″ | 44°51′60″ | SX37 | Xinjiang | 85°51′54″ | 47°32′53″ |
| SX05 | Xinjiang | 86°51′46″ | 44°20′28″ | SX38 | Xinjiang | 88°01′57″ | 47°14′16″ |
| SX06 | Xinjiang | 82°24′30″ | 44°49′48″ | SX39 | Xinjiang | 87°58′31″ | 47°18′39″ |
| SX07 | Xinjiang | 86°37′30″ | 44°28′59″ | SX40 | Xinjiang | 87°12′18″ | 44°07′12″ |
| SX08 | Xinjiang | 82°10′35″ | 44°37′59″ | SX41 | Xinjiang | 87°04′44″ | 42°15′58″ |
| SX09 | Xinjiang | 82°27′7″ | 44°37′51″ | SX42 | Xinjiang | 88°1′52 " | 47°14′16 " |
| SX01 | Xinjiang | 82°5′43″ | 44°39′25″ | SX43 | Xinjiang | 86°48′07″ | 40°10′06″ |
| SX10 | Xinjiang | 82°6′22″ | 44°53′2″ | SX44 | Xinjiang | 87°12′18″ | 44°07′12″ |
| SX11 | Xinjiang | 82°8′60″ | 44°39′1″ | SX45 | Xinjiang | 87°04′44″ | 42°15′58″ |
| SX12 | Xinjiang | 85°48′55″ | 41°43′36″ | SX46 | Xinjiang | 88°01′57″ | 47°14′16″ |
| SX13 | Xinjiang | 84°58′38″ | 44°34′17″ | SX47 | Xinjiang | 87°15′06″ | 44°08′31″ |
| SX14 | Xinjiang | 86°50′25″ | 44°23′50″ | SX48 | Xinjiang | 85°06′37″ | 44°19′03″ |

### 2.3. Sequence Alignment for SNP Calling and Quality Assessment

Taking into account the absence of a reference genome for *S. nigrum*, the SLAF reads of each sample were aligned to the *S. melongena* reference genome (ftp://ftp.solgenomics.net/genomes/Solanum_melongena_consortium/, accessed on 10 August 2020). A preliminary SLAF experiment using the genome of *S. melongena* was conducted. We called samples SNPs in the 66 *S. nigrum* individuals based on the corresponding genome-wide SLAF tags. The Genome Analysis [31] Toolkit and SAMtools [32] were used for SNP calling and the SNPs were quality-filtered with a minor allele frequency (MAF) $\geq 0.05$. Based on the above parameter, 616,533 SNP markers were retained for further analysis.

### 2.4. Genetic Differentiation Analyses

For the estimation of the level and pattern of genetic differentiation among geographic populations, we estimated fixation index Fst (F-statistics) values for the SNPs of all pairs of populations following Weir and Cockerham's method as implemented in the GENODIVE software v2.0 [33]. Diversity parameters, the value of the observed heterozygosity (Ho), and the expected heterozygosity (He) were employed using GENODIVE software v2.0.

**Table 2.** Summary statistics for specific-locus amplified fragment sequence data of *S. nigrum*.

| ID | Total Reads | GC Percentage (%) | Q30 Percentage (%) | ID | Total Reads | GC Percentage (%) | Q30 Percentage (%) |
|----|-------------|-------------------|--------------------|----|-------------|-------------------|--------------------|
| SB1 | 1,691,734 | 42.73 | 93.84 | SX15 | 1,650,570 | 43.41 | 94.14 |
| SH1 | 1,807,444 | 42.05 | 94.47 | SX16 | 1,519,690 | 43.44 | 94.39 |
| SH2 | 1,902,574 | 42.80 | 94.42 | SX17 | 1,737,036 | 42.87 | 93.68 |
| SH3 | 1,823,795 | 43.30 | 94.01 | SX18 | 2,002,105 | 43.27 | 94.15 |
| SH4 | 1,716,365 | 43.0 | 93.65 | SX19 | 1,736,208 | 43.40 | 94.31 |
| SJ1 | 2,662,305 | 42.15 | 94.57 | SX20 | 1,681,088 | 43.44 | 94.47 |
| SJ2 | 1,494,444 | 42.98 | 94.02 | SX21 | 1,622,367 | 43.09 | 94.48 |
| SJ3 | 1,735,455 | 43.30 | 94.68 | SX22 | 1,598,613 | 43.41 | 94.58 |
| SJ4 | 3,858,231 | 42.72 | 94.36 | SX23 | 1,534,640 | 43.47 | 94.78 |
| SJ5 | 1,643,277 | 43.10 | 93.62 | SX24 | 1,941,331 | 43.02 | 94.58 |
| SJ6 | 2,131,241 | 43.27 | 94.01 | SX25 | 2,100,514 | 42.83 | 94.29 |
| SJ7 | 1,588,142 | 42.71 | 93.70 | SX26 | 1,716,587 | 42.76 | 94.53 |
| SL1 | 1,883,192 | 43.04 | 94.02 | SX27 | 1,896,964 | 43.15 | 93.89 |
| SL2 | 1,616,090 | 43.44 | 94.53 | SX28 | 2,049,341 | 43.24 | 94.14 |
| SM1 | 1,657,653 | 42.95 | 94.42 | SX29 | 1,799,983 | 43.29 | 93.80 |
| SM2 | 2,166,560 | 42.33 | 93.63 | SX31 | 1,812,148 | 42.8 | 93.90 |
| SN1 | 2,109,751 | 42.85 | 93.91 | SX32 | 1,838,587 | 42.82 | 93.44 |
| SN2 | 1,676,914 | 43.00 | 94.43 | SX33 | 1,753,477 | 43.08 | 94.06 |
| SS1 | 1,620,348 | 42.88 | 94.09 | SX34 | 1,769,883 | 43.0 | 94.23 |
| SX01 | 2,450,367 | 41.97 | 94.76 | SX35 | 1,712,473 | 43.29 | 94.27 |
| SX02 | 1,723,640 | 42.68 | 93.92 | SX36 | 1,522,925 | 42.49 | 93.27 |
| SX03 | 2,213,391 | 41.92 | 94.75 | SX37 | 1,731,599 | 42.72 | 93.57 |
| SX04 | 1,973,084 | 42.03 | 94.79 | SX38 | 1,711,985 | 42.73 | 93.68 |
| SX05 | 1,958,204 | 42.30 | 94.77 | SX39 | 1,789,567 | 42.51 | 93.37 |
| SX06 | 2,121,610 | 42.05 | 94.63 | SX40 | 1,442,651 | 42.36 | 93.34 |
| SX07 | 2,053,106 | 42.13 | 94.73 | SX41 | 1,563,989 | 42.78 | 93.21 |
| SX08 | 2,393,941 | 41.88 | 94.58 | SX42 | 1,527,690 | 42.56 | 93.11 |
| SX09 | 2,205,426 | 42.09 | 94.63 | SX43 | 1,741,900 | 42.61 | 93.55 |
| SX10 | 2,023,837 | 42.51 | 94.57 | SX44 | 1,974,449 | 42.57 | 93.60 |
| SX11 | 2,404,368 | 42.03 | 94.49 | SX45 | 1,895,990 | 42.50 | 93.53 |
| SX12 | 1,750,841 | 42.12 | 94.55 | SX46 | 1,606,552 | 42.57 | 93.42 |
| SX13 | 1,863,520 | 42.87 | 94.41 | SX47 | 1,823,353 | 42.65 | 93.87 |
| SX14 | 1,451,126 | 43.35 | 94.15 | SX48 | 1,546,745 | 42.65 | 93.81 |

To estimates the evolutionary divergence between sequences, analyses were conducted using the Kimura 2-parameter method [34]. Evolutionary analyses were conducted in MEGA X [35]. This analysis involved 66 nucleotide sequences. All positions with less than 80% site coverage were eliminated, i.e., fewer than 20% alignment gaps, missing data, and ambiguous bases were allowed at any position (partial deletion option). There were a total of 15,527 positions in the final dataset.

*2.5. Phylogenetic Analysis*

We performed multiples analyses to examine the level of genetic diversity that exists within *S. nigrum* populations. All SLAF pair-end reads with clear index information were clustered on the basis of sequence similarity. Sequence similarity was detected using one-to-one alignment with BLAST [36]. A phylogenetic tree was constructed based on the SNPs using maximum likelihood analyses [37]. Neighbor-joining trees were reconstructed in MEGA X [35] using Nei's genetic distances between pairs of samples and 1000 bootstrap replicates. Population structure analysis was performed through ADMIXTURE based on a cross-validation error ranging from k = 1 to k = 10 [38,39]. In order to obtain insight into the gene flow and evolutionary relationships among populations, TreeMix models was used to construct maximum likelihood (ML) trees based on the allele frequency covariance matrix and added migration edges [40]. The genetic drift at genome-wide polymorphisms to infer

evolutionary relationships and directionality of gene flow between *S. nigrum* populations was evaluated.

## 3. Results

### 3.1. Phylogenetic Analyses on S. nigrum Populations

A total of 189,840 high-quality SLAFs was identified and used to call the SNPs with an average sequencing depth of 10.56-fold per individual. For each sample, the average sequencing depth ranged from $8.16\times$ fold to $14.84\times$ fold (Table 3). Of these, 90,590 were identified as polymorphic. Finally, 616,533 SNPs with a minor allele frequency (MAF) $\geq 0.05$ were selected for analyses of phylogenetic relationships and population structure after filtering based on SLAF-seq for the 66 samples. For each *S. nigrum* individual, the number of SNPs ranged from 266,597 to 450,693 (Table 4). The largest number of SNPs occurred on sample SJ4 (450,693 SNPs), followed by SX25 (364,002 SNPs), whereas the smallest number of SNPs was found on SX17 (266,597 SNPs).

**Table 3.** Summary statistics for specific-locus amplified fragment sequence data.

| ID | SLAF Number | Total Depth | Average Depth | ID | SLAF Number | Total Depth | Average Depth |
|---|---|---|---|---|---|---|---|
| SB1 | 143,008 | 1,397,095 | 9.77 | SX15 | 143,551 | 1,201,101 | 8.37 |
| SH1 | 139,623 | 1,558,808 | 11.16 | SX16 | 142,910 | 1,182,897 | 8.28 |
| SH2 | 150,756 | 1,514,661 | 10.05 | SX17 | 135,530 | 1,473,942 | 10.88 |
| SH3 | 148,052 | 1,447,856 | 9.78 | SX18 | 151,906 | 1,592,422 | 10.48 |
| SH4 | 144,849 | 1,383,746 | 9.55 | SX19 | 143,270 | 1,344,168 | 9.38 |
| SJ1 | 143,453 | 2,354,446 | 16.41 | SX20 | 144,038 | 1,352,983 | 9.39 |
| SJ2 | 139,706 | 1,221,842 | 8.75 | SX21 | 144,141 | 1,298,644 | 9.01 |
| SJ3 | 148,056 | 1,347,424 | 9.10 | SX22 | 145,000 | 1,224,795 | 8.45 |
| SJ4 | 171,626 | 3,081,308 | 17.95 | SX23 | 142,066 | 1,189,055 | 8.37 |
| SJ5 | 146,279 | 1,274,606 | 8.71 | SX24 | 150,371 | 1,563,158 | 10.40 |
| SJ6 | 153,773 | 1,702,217 | 11.07 | SX25 | 153,599 | 1,678,791 | 10.93 |
| SJ7 | 139,864 | 1,326,283 | 9.48 | SX26 | 148,364 | 1,362,955 | 9.19 |
| SL1 | 150,344 | 1,497,548 | 9.96 | SX27 | 151,134 | 1,479,202 | 9.79 |
| SL2 | 142,746 | 1,290,494 | 9.04 | SX28 | 152,221 | 1,635,532 | 10.74 |
| SM1 | 144,398 | 1,334,945 | 9.24 | SX29 | 145,849 | 1,442,431 | 9.89 |
| SM2 | 139,926 | 1,877,997 | 13.42 | SX31 | 146,580 | 1,511,018 | 10.31 |
| SN1 | 142,636 | 1,782,163 | 12.49 | SX32 | 147,503 | 1,509,826 | 10.24 |
| SN2 | 146,239 | 1,327,197 | 9.08 | SX33 | 148,484 | 1,394,929 | 9.39 |
| SS1 | 134,045 | 1,351,964 | 10.09 | SX34 | 147,378 | 1,434,272 | 9.73 |
| SX01 | 145,563 | 2,152,103 | 14.78 | SX35 | 146,085 | 1,356,216 | 9.28 |
| SX02 | 136,493 | 1,474,738 | 10.80 | SX36 | 139,212 | 1,215,151 | 8.73 |
| SX03 | 140,107 | 1,955,306 | 13.96 | SX37 | 141,906 | 1,452,165 | 10.23 |
| SX04 | 137,242 | 1,734,153 | 12.64 | SX38 | 142,509 | 1,418,841 | 9.96 |
| SX05 | 142,744 | 1,662,479 | 11.65 | SX39 | 144,886 | 1,457,086 | 10.06 |
| SX06 | 141,570 | 1,858,567 | 13.13 | SX40 | 135,976 | 1,186,188 | 8.72 |
| SX07 | 140,762 | 1,798,000 | 12.77 | SX41 | 106,088 | 1,140,162 | 10.75 |
| SX08 | 142,846 | 2,119,727 | 14.84 | SX42 | 137,456 | 1,246,916 | 9.07 |
| SX09 | 145,038 | 1,910,647 | 13.17 | SX43 | 141,769 | 1,457,711 | 10.28 |
| SX10 | 147,941 | 1,680,924 | 11.36 | SX44 | 144,564 | 1,689,305 | 11.69 |
| SX11 | 150,815 | 2,067,161 | 13.71 | SX45 | 144,316 | 1,591,007 | 11.02 |
| SX12 | 137,667 | 1,510,621 | 10.97 | SX46 | 137,927 | 1,350,467 | 9.79 |
| SX13 | 151,122 | 1,475,451 | 9.76 | SX47 | 146,675 | 1,493,755 | 10.18 |
| SX14 | 139,948 | 1,141,327 | 8.16 | SX48 | 138,873 | 1,285,432 | 9.26 |

**Table 4.** Summary statistics for SNP sequence data.

| ID | SNP Number | Hetloci Ratio (%) | Integrity Ratio (%) | ID | SNP Number | Hetloci Ratio (%) | Integrity Ratio (%) |
|---|---|---|---|---|---|---|---|
| SB1 | 300,918 | 17.91 | 48.80 | SX15 | 340,669 | 19.00 | 55.25 |
| SH1 | 279,620 | 17.93 | 45.35 | SX16 | 334,821 | 18.15 | 54.30 |
| SH2 | 351,232 | 19.60 | 56.96 | SX17 | 266,597 | 16.87 | 43.24 |
| SH3 | 343,657 | 19.14 | 55.74 | SX18 | 355,004 | 19.54 | 57.58 |
| SH4 | 317,807 | 17.97 | 51.54 | SX19 | 327,964 | 18.31 | 53.19 |
| SJ1 | 276,991 | 18.88 | 44.92 | SX20 | 322,766 | 17.95 | 52.35 |
| SJ2 | 297,219 | 17.30 | 48.20 | SX21 | 323,363 | 17.86 | 52.44 |
| SJ3 | 353,923 | 19.30 | 57.40 | SX22 | 345,938 | 18.78 | 56.11 |
| SJ4 | 450,693 | 28.73 | 73.10 | SX23 | 336,435 | 18.48 | 54.56 |
| SJ5 | 341,109 | 18.93 | 55.32 | SX24 | 346,155 | 19.22 | 56.14 |
| SJ6 | 359,735 | 19.94 | 58.34 | SX25 | 364,002 | 20.11 | 59.04 |
| SJ7 | 284,537 | 17.15 | 46.15 | SX26 | 343,732 | 19.22 | 55.75 |
| SL1 | 349,578 | 19.69 | 56.70 | SX27 | 357,266 | 19.79 | 57.94 |
| SL2 | 321,662 | 17.92 | 52.17 | SX28 | 354,589 | 19.76 | 57.51 |
| SM1 | 321,841 | 17.88 | 52.20 | SX29 | 329,492 | 18.55 | 53.44 |
| SM2 | 268,565 | 17.91 | 43.56 | SX31 | 310,206 | 18.43 | 50.31 |
| SN1 | 295,616 | 18.08 | 47.94 | SX32 | 318,428 | 18.67 | 51.64 |
| SN2 | 334,074 | 18.23 | 54.18 | SX33 | 337,137 | 18.91 | 54.68 |
| SS1 | 274,926 | 16.70 | 44.59 | SX34 | 326,911 | 18.22 | 53.02 |
| SX01 | 295,832 | 19.07 | 47.98 | SX35 | 332,924 | 18.43 | 53.99 |
| SX02 | 267,309 | 17.33 | 43.35 | SX36 | 301,753 | 17.80 | 48.94 |
| SX03 | 272,426 | 18.23 | 44.18 | SX37 | 287,012 | 17.57 | 46.55 |
| SX04 | 267,987 | 17.77 | 43.46 | SX38 | 301,108 | 17.77 | 48.83 |
| SX05 | 300,601 | 17.99 | 48.75 | SX39 | 315,850 | 18.54 | 51.23 |
| SX06 | 279,522 | 18.32 | 45.33 | SX40 | 282,203 | 17.29 | 45.77 |
| SX07 | 279,250 | 18.17 | 45.29 | SX41 | 302,783 | 8.39 | 49.11 |
| SX08 | 279,198 | 18.55 | 45.28 | SX42 | 290,009 | 17.36 | 47.03 |
| SX09 | 297,579 | 18.89 | 48.26 | SX43 | 287,487 | 17.25 | 46.62 |
| SX10 | 323,345 | 19.13 | 52.44 | SX44 | 290,581 | 17.74 | 47.13 |
| SX11 | 314,870 | 19.83 | 51.07 | SX45 | 300,420 | 18.20 | 48.72 |
| SX12 | 271,604 | 17.61 | 44.05 | SX46 | 279,457 | 17.04 | 45.32 |
| SX13 | 355,317 | 19.87 | 57.63 | SX47 | 315,801 | 18.34 | 51.22 |
| SX14 | 317,163 | 17.48 | 51.44 | SX48 | 285,131 | 17.00 | 46.24 |

*3.2. Genetic Differences and Genetic Distances of S. nigrum*

After SNP filtering, a genetic diversity study was carried out using 616,533 SNPs within 66 *S. nigrum* samples. The genetic diversity analysis showed that mean effective allele number (Ne) was 1.81. Observed (Ho) heterozygosity number ranged from 0.507 to 0.7403, with an average 0.6306. The expected (He) heterozygosity varied from 0.4048 to 0.4597 with an average 0.434. The average polymorphic information content value was 0.3353. The mean Shannon diversity index was 0.621 varied from 0.5865 to 0.6479. The average MAF was 0.3782, ranged from 0.3388 to 0.4194. Nei diversity index 0.419 to 0.613, average was 0.5019. These estimations indicated that there is a considerable amount of genetic variation among each samples.

The analysis of genetic differentiation showed that there was a low level of genetic differentiation (Fst < 0.1000) among each geographical population (Table 5). The pairwise genetic distance values among all samples ranged from 0.001 to 0.573, with an average of 0.308 (Supplementary Table S1). The minimum distance was found between SX27 and SX48 collected from two nearby locations in Xinjiang. The largest distance was observed between SJ1 and SX33, collected from the northeast (123°43′53″ E, 44°10′83″ N) and northwest (82°06′20″ E, 44°58′02″ N) of China, respectively.

**Table 5.** Estimates of genetic differentiation (Fst) over sequence pairs and between geographical populations.

| Population | HLJ | IM | JL | LN | HN | XJ |
|---|---|---|---|---|---|---|
| HLJ | 0.009 | | | | | |
| IM | | | | | | |
| JL | 0.007 | 0.005 | | | | |
| LN | 0.005 | 0.039 | 0.005 | | | |
| HN | 0.036 | 0.007 | 0.038 | 0.047 | | |
| XJ | 0.001 | 0.012 | 0.005 | 0.003 | 0.020 | |

Population name HLJ: Heilongjiang, HN: Henan, IM: Inner Mongolia; JL: Jilin; LN: Liaoning; XJ: Xinjiang.

### 3.3. Population Structure of S. nigrum

The ML (maximum likelihood) phylogenetic tree (Figure 1) showed strong evidence for two clusters: cluster A was a Xinjiang group; cluster B was a widespread group with individuals extending from northwest to northeast China. We found some correlation between geographic origin and genetic structure among 66 individuals. In cluster B, northeast populations (JL, LN, IN) could be clustered together roughly, whereas most *S. nigrum* individuals from Xinjiang could be clustered into a subgroup (Figure 1). These two subclusters were well separated and therefore distantly related.

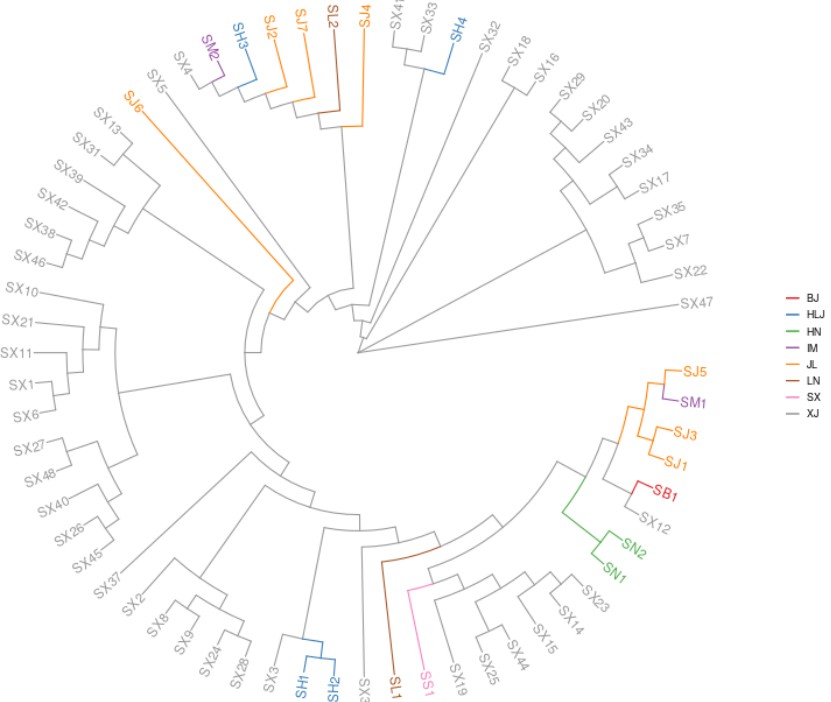

**Figure 1.** Phylogenetic tree constructed according to the neighbor-joining method with polymorphic SNPs based on 66 S. *nigrum* individuals. The colors of branches indicate genotypes corresponding to different sub-origin sets. BJ: Beijing; HLJ: Heilongjiang, HN: Henan, IM: Inner Mongolia; JL: Jilin; LN: Liaoning; SX: Shaanxi; XJ: Xinjiang.

To further understand the evolutionary history of *S. nigrum*, we used a Bayesian clustering algorithm with admixed models [41] to estimate the ancestral proportions for each individual. The population structure analysis was performed through ADMIXTURE based on a cross-validation error ranging from k = 1 to k = 10. The value of error ranged from 0.479 to 0.698. The lowest cross-validation error was observed at k = 3 (Figure 2). This suggests that the 66 *S. nigrum* samples could have originated from three different genetic clusters (Q1, Q2, and Q3) (Figure 2). Of three clusters, cluster II (Q2) comprised the most

germplasms with 34 samples, followed by cluster I (Q1) (18 samples) and cluster III (Q3) (14 samples). The Xinjiang site was the only location where all genetic clusters were found. A high level of admixture was detected in the Xinjiang populations. The populations from Xinjiang were distributed in three clusters, suggesting these populations were genetically diverse (Figure 3).

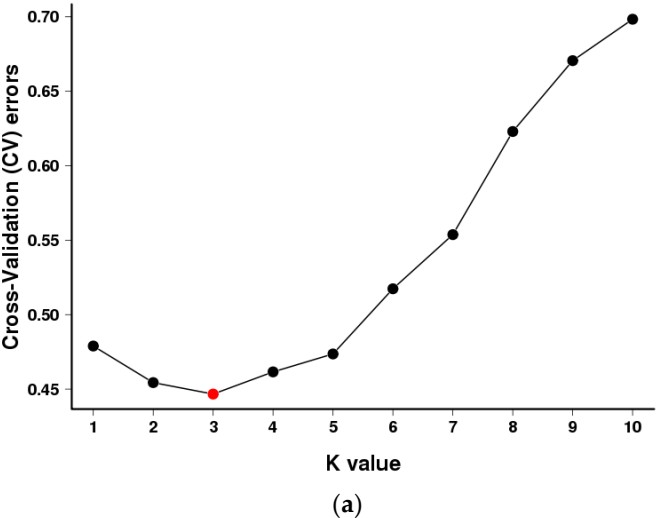

(**a**)

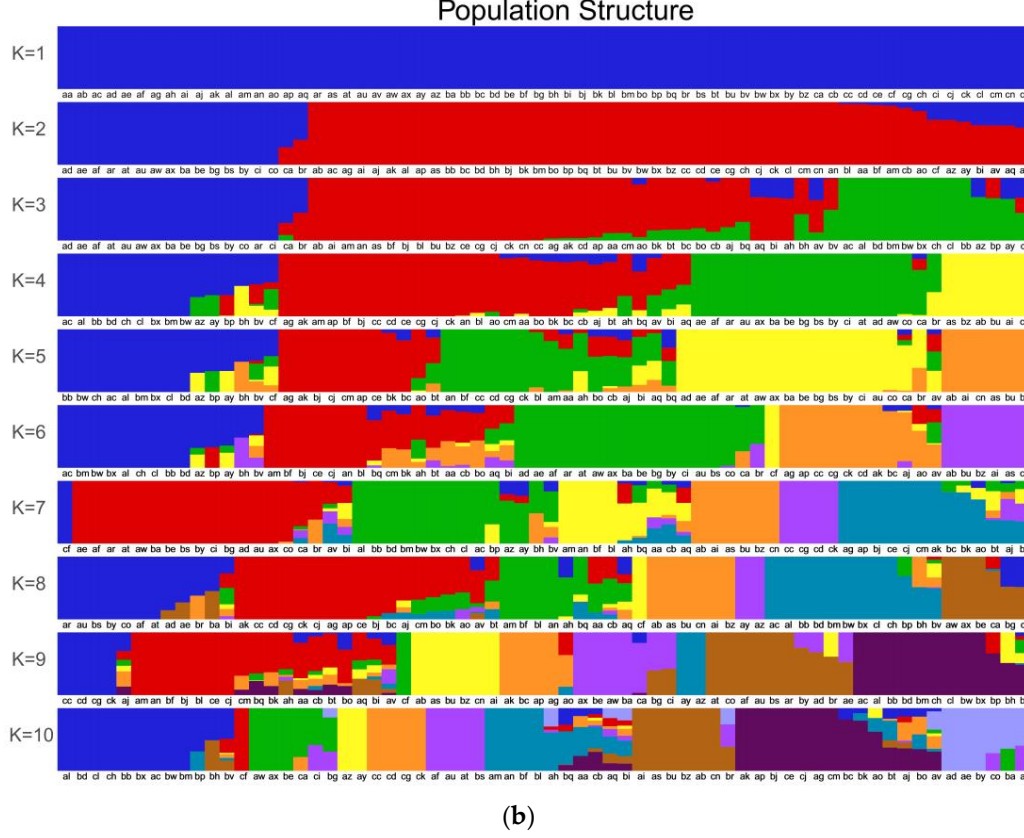

(**b**)

**Figure 2.** Population structure analysis of *S. nigrum* with the ADMIXTURE program using genome-wide SNP markers (*n* = 616,533). (**a**) The estimated cross-validation errors for different grouping results (K value); (**b**) The individuals were divided into three subgroups (there was a minimum K-value when K = 3), within each subgroup, the individuals were ordered according to the genetic component, and each line gives the sub-group value, with each accession shown as a vertical line partitioned into K colored components representing inferred membership in K genetic clusters.

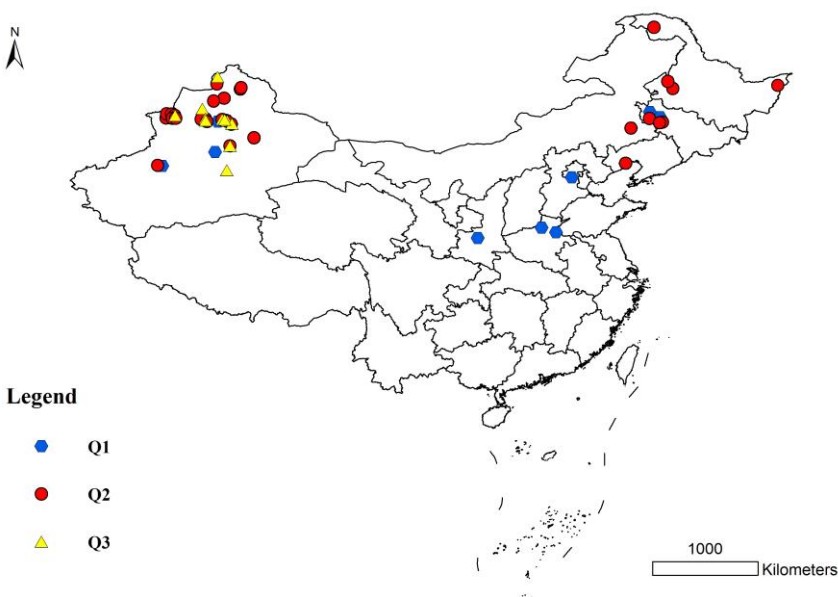

**Figure 3.** Distribution of *S. nigrum* used in this study and the different ancestors.

The first genetic cluster (Q1) was essentially represented by individual samples originating from Xinjiang, Beijing, Jilin, Inner Mongolia, Henan, and Shaanxi. This ancestry ranged from 0.557 to 0.999. The samples from SL1 and SX36 in Q1 had an average ancestry proportion for their major cluster below 0.75. The second genetic cluster was mainly represented by samples originating from Heilongjiang, Jilin, Liaoning, Inner Mongolia, and Xinjiang. The ancestry proportion within these populations ranged from 0.501 to 0.999. Three individuals from Heilongjiang, three samples from Xinjiang, and one from Jilin in Q2 had an ancestry proportion below 0.75. The last genetic cluster Q3 was only detected within samples collected in Xinjiang. The ancestry ranged from 0.440 to 0.999. Four samples in Q3 had an ancestry proportion below 0.75.

The geographical distances were not significantly correlated with the genetic distances among samples. Although the greatest geographic distance was between the populations from the SH2 (133°52′53″ E, 47°24′19″ N) and SX24 (80°54′50″ E, 40°34′57″ N) sites, these two samples were found to belong to the same cluster (Q2). However, the samples SM1 and SM2 from Inner Mongolia collected in very close sites were observed to belong to Q1 and Q2, respectively. The genetic distance between SM1 and SM2 was 0.428 (Supplementary Table S1), indicated that there was a significant difference between SM1 and SM2. Other samples, including SX41 and SX45, SX4 and SX44, SX13 and SX17, SX20 and SX36, and SX16 and SX14, all pairs of samples collected from very close sites, were observed to belong to different clusters (Figure 3).

To understand the history of divergence and admixture, we used TreeMix for the six geographical groups including the samples from Xinjiang, Liaoning, Heilongjiang, Jilin, Henan, and Inner Mongolia. In the TreeMix analysis (Figure 4), there was a significant gene flow event that could be found among the *S. nigrum* populations, which indicated that extensive gene flow had occurred in northeastern and western China. Between the two clusters, extensive gene flow from Liaoning (LN) to Heilongjiang (HLJ) was observed. The Inner Mongolia (IM) group showed more significant genetic similarity with the Liaoning group that differed markedly from the other groups.

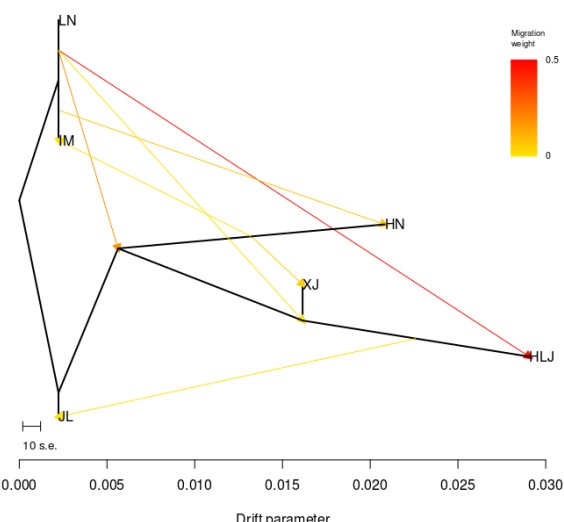

**Figure 4.** TreeMix analysis of the six geographic populations. The arrow corresponds to the direction of migration. Seven migration edges were allowed, as inferred using OptM. The migration edges are colored according to their weight ($\hat{w}$). The scale bar indicates ten times the average standard error of the values in the covariance matrix. LN: Liaoning population; IM: Inner Mongolia population; JL: Jilin population; XJ: Xinjiang population; HLJ: Heilongjiang population; HN: Henan population.

## 4. Discussion

*S. nigrum* is one of the most common and difficult-to-control weeds in many crops, including cotton, soybean, and sugar beet, due to biological characteristics that enhance the adaptive ability of the weed in a wide range of environments. The success of weeds in the ecosystem can be attributed to genetic diversity [42]. An effective weed management program should be based upon the diversity level of each weed species [43]. In the present study, using SLAF-seq technology, 66 *S. nigrum* individuals from a wide range in China were collected and genome-wide SNPs were obtained. The genomic data provide us novel insight into *S. nigrum*'s genetic diversity. Single-nucleotide polymorphisms are much more abundant than other molecular markers in the genome. In a previous study of 14 *Solanum* accessions, 130 polymorphic bands were identified based on ISSR [19,44]. In this study, a total of 189,840 SLAFs was detected from 66 individuals of *S. nigrum*, and almost half of the SLAF markers were polymorphic, which also suggests that the observed genetic diversity of *S. nigrum* in China is rich.

It is indicated that *S. nigrum* populations have high genetic diversity based on the observed heterozygosity (Ho > 0.5). However, the pairwise Fst comparisons showed a low level of genetic differentiation among *S. nigrum* populations. Admixture and low genetic differentiation among populations may be caused by high gene flow [45]. A high degree of gene flow can increase genetic diversity and can have significant homogenizing effects [46]. In this study we found extensive gene flow and admixture between northeast to northwest populations. This suggests that environmental factors influence the genetic diversity of *S. nigrum*.

Population structure is the result of both present and historical processes, and many factors may change the geographical distributions of plant species [47]. On the basis of the phylogenetic analysis, total 14 individuals from the Xinjiang population clustered together without other individuals, suggesting the origin of the Q3 population differs from that of the other populations. The genetic structural analysis revealed that samples from Xinjiang comprise three genotypes, which is in accordance with the phylogenetic results. Therefore the Xinjiang site could be viewed as a repository of genetic diversity.

This study highlights the value of the de novo sequencing of *S. nigrum* genomes to study the genomic patterns in a phylogenetic framework. The SLAF-seq analysis of various *S. nigrum* samples revealed the genetic diversity, which was consistent with the

results of phylogenomic analyses. Before the current study, very little was known about the population genetic structure of *S. nigrum* in China. The present results increase our understanding of *S. nigrum*'s genetic structure and diversity. The high rate of heterozygosity in every population suggested that migration of *S. nigrum* accelerated the gene flow and genetic evolution, which may facilitate *S. nigrum* to adapt to their environment.

The data presented herein may represent the basis for future studies related to *S. nigrum* genetic evolution.

## 5. Conclusions

In summary, we herein reveal the genetic diversity of *S. nigrum* at the population and species level. Considerable genetic differentiation was observed among individuals. A phylogenetic tree suggested the relationship between many individuals was not inconsistent with the geographical location. All individuals were clustered into three genotypes according to a structural analysis. These data provided important information about *S. nigrum* phylogenomics and will be useful for further understanding of the evolutionary genetics of weed adaptation to the agricultural environment.

**Supplementary Materials:** The following supporting information can be downloaded at: https://www.mdpi.com/article/10.3390/agronomy13030832/s1, Table S1: Values of pairwaise genetic distance among *S. nigrum* populations.

**Author Contributions:** J.L. and S.W. performed data analysis and drafted the manuscript; Y.Z. (Yixiao Zhang), Z.M., and L.L. participated in the experiments; Z.H. and Y.Z. (Yuyong Zhu) provided some samples; H.H. conceived and designed the experiments and performed data analysis. All authors have read and agreed to the published version of the manuscript.

**Funding:** This research was funded by the project of the Key R&D program of Xinjiang Uygur Autonomous Region: 2022B02043-3; the Key Research and Development Program of Special Funds for Construction Corps: 2018AA006; and the Beijing Natural Science Foundation: 6212027, 6182033. No conflicts of interest have been declared.

**Data Availability Statement:** Data available from the author.

**Conflicts of Interest:** The authors declare no conflict of interest.

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
