# Peer review of "Genetic Diversity and Population Structure in Solanum nigrum Based on Single-Nucleotide Polymorphism (SNP) Markers"

_agronomy, doi:10.3390/agronomy13030832_

Round 1

Reviewer 1 Report

The manuscript of Li et al. is well presented and english is easy to read. The methodology is well explained and the results well analyzed. Despite the many results presented, I feel that questions remain regarding the purpose of this study.

 Besides some modifications need to be taken into account as follows:

 In the title use "Genetic diversity" instead of "Gene diversity" and 'based' instead of 'base'

 Abstract section:

Line 16: delete "single-nucleotide polymorphisms" and let SNPs as you already mentioned this in line 15.  

 Introduction section:

Line 40 'nigrum' instead of 'nibrum'

Line 76: “This is the first to use SNPs” something is missing here, verify!!! the first “report” maybe !!!

Line 77: add a sentence at the end showing why you want to see if diversity is randomly distributed or has a patterned spatial distribution. Please clariy your objectifs.

 Material & Methods section:

In plant material (lines 80-87): add pictures if you have of these S. nigrum populations

Line 90: how many grams (for each sample) you used of fresh young leave you used to extract DNA

Line 99: define this quality control 'Q30' to be ubderstood by readers

line 113: you should define the fixation index (Fst) and what information will be provided once estimating this Fst.

line 117-'the' instead of 'of'

line 127: for Blast add reference or link of the  webpage you used.

line 140: please correct, the largest SNP occured in SJ4 sample

Line 146: S. nigrum in itallic, verify in all the manuscript

line 154: you should add a table in supplementary material showing genetic differentiation (Fst) among the different geographical population. A Table showing pairwaise genetic distance as well.

Line 162: give the hole words of the abreviatio ML

Figure 1: you should enhance the quality of this figure, it is not clear the legend neither the name of genotypes. Same for Figure 2, it should be enhanced to be easy to read it.

Line 212: when you talk about genetic distance, a Table should be added to see the data. You can added in Supplementary material if not possible in the main text.

Line 223: how did you determined or measured the gene flow, add a sentence in material and methods section.

Line 224: Please add after Liaoning (LN). You should add the abreviation of each location in lines 224 to 225.

Discussion section

Line 252: Homogenizing effect ?? what you mean

Line 279: "will be useful for further research", You should specify this !!

Author Response

Response to Reviewer 1 Comments

Thank you for your letter of and for the referee’s comments concerning our manuscript entitled “Genetic diversity and population structure in Solanum nigrum based on single-nucleotide polymorphism (SNP) markers”. We have studied their comments carefully and have made correction which we hope meet with their approval.

The manuscript has been revised according to reviewers’ suggestions and comments, and detailed specifications of our revision are given in the followings:

Point 1: In the title use "Genetic diversity" instead of "Gene diversity" and 'based' instead of 'base'

Response 1: we were really sorry for our careless mistakes. Thank you for your reminder. Now the name of the title is “Genetic diversity and population structure in Solanum nigrum based on single-nucleotide polymorphism (SNP) markers”

Point 2: Line 16: delete "single-nucleotide polymorphisms" and let SNPs as you already mentioned this in line 15. 

Response 2: Thanks for your suggestion. We have deleted accordingly.

Point 3: Line 40 'nigrum' instead of 'nibrum'

Response 3: Thank you for pointing this out. We have corrected the “nibrum” into “nigrum”.

Point 4: Line 76: “This is the first to use SNPs” something is missing here, verify!!! the first “report” maybe !!!

Response 4: I am so sorry for this mistake. We have corrected according your suggestion.

Point 5: Line 77: add a sentence at the end showing why you want to see if diversity is randomly distributed or has a patterned spatial distribution. Please clariy your objectifs.

Response 5: Thanks for your good suggestion. We have revised the introduction to make the objectifs of study more clearer.

Point 6: In plant material (lines 80-87): add pictures if you have of these S. nigrum populations

Response 6: I am so sorry, there are not pictures of these S. nigrum populations.

Point 7: how many grams (for each sample) you used of fresh young leave you used to extract DNA

Response 7: For each sample, about 100mg fresh young leave were used to extract DNA. The information has been supplyed.

Point 8: Line 99: define this quality control 'Q30' to be ubderstood by readers

Response 8: Thanks for your suggestion. We have define the Q30 in revised manuscript.

Point 9: line 113: you should define the fixation index (Fst) and what information will be provided once estimating this Fst.

Response 9: Thanks a lot. We have define the Fst and provided the information.

Point 10: line 117-'the' instead of 'of'

Response 10: Thanks for your suggestion. We have corrected “of” into “the”.

Point 11: line 127: for Blast add reference or link of the webpage you used.

Response 11: Thanks a lot for your carefulness. The reference of Blast has been listed in the revised manuscript.

Point 12: line 140: please correct, the largest SNP occured in SJ4 sample

Response 12: I am so sorry for this mistake. Now the “SJ4” has instead of “JL4”.

Point 13: Line 146: S. nigrum in itallic, verify in all the manuscript

Response 13: Thanks a lot for your carefulness. We have carefully checked the manuscript and corrected the errors accordingly.

Point 14: line 154: you should add a table in supplementary material showing genetic differentiation (Fst) among the different geographical population. A Table showing pairwaise genetic distance as well.

Response 14: Thank you very much for the suggestion. I have added table of Fst in revised manscript and a table in supplementary material showing pairwaise genetic distance.

Point 15: Line 162: give the hole words of the abreviatio ML

Response 15: The hole words of ML has been added in manuscript now.

Point 16: Figure 1: you should enhance the quality of this figure, it is not clear the legend neither the name of genotypes. Same for Figure 2, it should be enhanced to be easy to read it.

Response 16: Thanks a lot for your good suggestions. We have enhance the quality of the figure 1 and figure 2.

Point 17: Line 212: when you talk about genetic distance, a Table should be added to see the data. You can added in Supplementary material if not possible in the main text.

Response 17: A table showing pairwaise genetic distance has been added in supplementary material.

Point 18: Line 223: how did you determined or measured the gene flow, add a sentence in material and methods section.

Response 18: We have add the method of gene flow in material and methods section.

Point 19: Line 224: Please add after Liaoning (LN). You should add the abreviation of each location in lines 224 to 225.

Response 19:  We have add abreviation of each location in these lines.

Point 20: Line 252: Homogenizing effect ?? what you mean

Response 20: We cite the reference (Wards. Genetic analysis of invasive plant populations at different spatial scales. Biol Invasions, 2006, 8, 541–552.) to discuss the relationship between the genetic differentiation and gene flow among S. nigrum populations. The lower level of genetic differentiation indicated that extensive gene flow between S. nigrum populations might lead to the homogenizing effect of gene flow at spatial scales.    

Point 21: Line 279: "will be useful for further research", You should specify this !!

Response 21: These data provided important information about S. nigrum phylogenomics and will be useful for further research understanding of the evolutionary genetics of weed adaptation to agricultural environment.

Reviewer 2 Report

The manuscript is of wrathful consideration for weed scientists and agronomists. The write-up of whole manuscript is up to mark. There are several suggestions that need to be incorporated before final decision as elaborated below:

-          The Abstract needs few details about materials and methodology of the study.

-          Please note the opening paragraph of the introduction could provide stronger context to the paper, and, similarly, the findings at the end could potentially be richer.

-          At the end of Introduction section, there should be clear hypothesis and objectives of the designed study.

-          Methods of the study and also statistical analysis are confusing.

-          The Results and Discussion section is okay.

-          A cutting conclusion section is needed.

Author Response

Response to Reviewer 1 Comments

  Thank you for your comments concerning our manuscript entitled “Genetic diversity and population structure in Solanum nigrum based on single-nucleotide polymorphism (SNP) markers”.  Those comments are all valuable and very helpful for revising and improving our paper. We have studied comments carefuly and have made correction which we hope meet with approval. We tried our best to improve the manuscript and made some changes in the manuscript. Revised portion are marked in red in the paper. The main corrections in the paper and the responds to your comments are as flowing:

Point 1: The Abstract needs few details about materials and methodology of the study.

Response 1: Thanks for your suggestion. We have few some details about materials and methodology of Abstract.

Point 2: Please note the opening paragraph of the introduction could provide stronger context to the paper, and, similarly, the findings at the end could potentially be richer.

Response 2: We have revised the introduction to provide more background context of S. nigrum. We hope the introduction could meet with their approval now.

Point 3: At the end of Introduction section, there should be clear hypothesis and objectives of the designed study.

Response 3: Thanks for your good suggestion. We have add the hypothesis and objectives of the study at the end of introduction section now.

Point 4: Methods of the study and also statistical analysis are confusing.

Response 4: Thanks for your comment. Methods of the study has been improved now.

Point 5: A cutting conclusion section is needed.

Response 5: Thanks for your reminding. A cutting conclusin section has been added at the end of the manuscript.